# Unveiling Non-Covalent Interactions in Novel Cooperative Photoredox Systems for Efficient Alkene Oxidation in Water

**DOI:** 10.3390/molecules29102378

**Published:** 2024-05-18

**Authors:** Isabel Guerrero, Clara Viñas, Francesc Teixidor, Isabel Romero

**Affiliations:** 1Institut de Ciencia de Materials de Barcelona (ICMAB-CSIC), Campus UAB, E-08193 Bellaterra, Spain; iguerrero@icmab.es (I.G.); clara@icmab.es (C.V.); 2Departament de Química and Serveis Tècnics de Recerca, Universitat de Girona, C/M. Aurèlia Campmany, 69, E-17003 Girona, Spain

**Keywords:** photoredox alkene oxidation, metallacarborane, ruthenium, ion-pair cooperative catalysis, aqueous medium

## Abstract

A new cooperative photoredox catalytic system, [Ru^II^(trpy)(bpy)(H_2_O)][3,3′-Co(8,9,12-Cl_3_-1,2-C_2_B_9_H_8_)_2_]_2_, **5**, has been synthesized and fully characterized for the first time. In this system, the photoredox catalyst [3,3′-Co(8,9,12-Cl_3_-1,2-C_2_B_9_H_8_)_2_]^−^ **[Cl_6_-1]**^−^, a metallacarborane, and the oxidation catalyst [Ru^II^(trpy)(bpy)(H_2_O)]^2+^, **2** are linked by non-covalent interactions. This compound, along with the one previously synthesized by us, [Ru^II^(trpy)(bpy)(H_2_O)][(3,3′-Co(1,2-C_2_B_9_H_11_)_2_]_2_, 4, are the only examples of cooperative molecular photocatalysts in which the catalyst and photosensitizer are not linked by covalent bonds. Both cooperative systems have proven to be efficient photocatalysts for the oxidation of alkenes in water through Proton Coupled Electron Transfer processes (PCETs). Using 0.05 mol% of catalyst **4**, total conversion values were achieved after 15 min with moderate selectivity for the corresponding epoxides, which decreases with reaction time, along with the TON values. However, with 0.005 mol% of catalyst, the conversion values are lower, but the selectivity and TON values are higher. This occurs simultaneously with an increase in the amount of the corresponding diol for most of the substrates studied. Photocatalyst **4** acts as a photocatalyst in both the epoxidation of alkenes and their hydroxylation in aqueous medium. The hybrid system **5** shows generally higher conversion values at low loads compared to those obtained with **4** for most of the substrates studied. However, the selectivity values for the corresponding epoxides are lower even after 15 min of reaction. This is likely due to the enhanced oxidizing capacity of Co^IV^ in catalyst **5**, resulting from the presence of more electron-withdrawing substituents on the metallacarborane platform.

## 1. Introduction

Epoxides are an important and versatile part of intermediate and basic components that are used to obtain more elaborate chemical products in both organic synthesis and in the industrial production of fine and bulk chemical products [1,2]. For example, epoxides can be transformed into a variety of functionalized products as diols, aminoalcohols, allylic alcohols, ketones, etc. [3,4,5,6]. The chemical oxidation of alkenes to obtain the corresponding epoxides has been carried out using different metal catalysts such as Mn-salen catalyst and NaClO or variations of these [7,8], Ti(*i*PrO)_4_ [9,10], and Ru-aqua complexes [11,12], among others. Most of these processes have been studied in organic media.

The development of photocatalytic methods and systems for organic transformations is challenging given the significant environmental and economic impact that this entails [13]. Cooperative photoredox catalysis represents a significant advancement in this field. It involves two catalysts: one that is photochemically active and another that is redox active, even in the absence of light [14]. The redox-active metal complex serves as a catalyst to activate either a water molecule or an organic substrate through a proton-coupled electron transfer (PCET) mechanism. Currently, photochemical systems studied for the oxidation of substrates involve a photocatalyst [15,16,17] or a photocatalyst combined with a transition metal catalyst based on polypyridyl compound [18,19,20,21]. However, a few examples of photochemical epoxidation of alkenes have been carried out with Earth-abundant metals [22,23,24]. Manganese and cobalt compounds all containing the carboranylcarboxylate ligand, [1-CH_3_-2-CO_2_-1,2-*closo-*C_2_B_10_H_10_]^−^, have been tested in the epoxidation of aliphatic and aromatic alkenes using peracetic acid as the oxidant [25]. The catalytic results highlight the role of the carboranylcarboxylate ligand in the selectivity of the processes, and it was found that coordination of the carboranylcarboxylate ligand to the metal ions is key to their catalytic performance.

On the other hand, it is well known that some boron clusters interact with light [26,27,28] and that they have been studied as catalysts in different processes. The most well-known metallacarborane is the anionic cobaltabis(dicarbollide) [29] that can be synthesized in high yield by a fast and environmentally friendly solid-state reaction [30]. The sandwich compound Na [3,3′-Co(1,2-C_2_B_9_H_11_)_2_], **Na [1]**, found by Fuentes et al. [31], has many possibilities to form hydrogen bonds (e.g., C_c_-H···O or C_c_-H···X (X = halogen) as well as dihydrogen bonding C_c_-H···H-B and B-H···H-N (C_c_ stands for the cluster carbon atoms)) [32,33,34]. **Na [1]** is highly stable in water, but even at low concentrations it forms aggregates (vesicles and micelles) [35,36,37] and can form ion-pair complexes through hydrogen and dihydrogen interactions. These supramolecular interactions appear to be significant in electron transfer processes and therefore in the performance and efficiency of photocatalytic systems.

Recently, we have shown that Na [1] and its dichloro (**Na**[**Cl_2_-1**]) and hexachloro (Na[**Cl_6_-1**]), acting both as catalyst and photosensitizer, have been highly efficient in the photooxidation of alcohol [38] and Na [1] in the oxidation of alkenes [24] in water, through single electron transfer processes (SET). We have also supported the metallabis(dicarbollide) catalyst on silica-coated magnetite nanoparticles [39]. This system has proven to be a green and sustainable heterogeneous catalytic system, highly efficient, and easily reusable for the photooxidation of alcohols in water. Finally, we have synthesized a cooperative system where the cobaltabis(dicarbollide) was linked by non-covalent interactions to a redox active oxidation catalyst that represented an efficient photocatalyst for the photooxidation of alcohols in water through PCET [40]. This specific cooperative photocatalytic system has not yet been explored for the photooxidation of alkenes in an aqueous environment.

With all this in mind, we describe here the synthesis of a new ruthenium-cobaltabis(dicarbollide) compound, [Ru^II^(terpy)(bpy)(H_2_O)][3,3′-Co(8,9,12-Cl_3_-1,2-C_2_B_9_H_8_)_2_]_2_
**5** (see [Fig molecules-29-02378-ch001]), where the **[**Ru**-**OH_2_**]** cation belongs to the family of redox oxidation catalysts [41] and the anion is the hexachloro cobaltabis(dicarbollide), [3,3′-Co(8,9,12-Cl_3_-1,2-C_2_B_9_H_8_)_2_]^−^, **[Cl_6_-1]^−^**. Also, their complete spectroscopic and electrochemical characterization has been conducted. The photocatalytic behavior of **5**, together with that of the previously synthetized [Ru^II^(terpy)(bpy)(H_2_O)][(3,3′-Co(1,2-C_2_B_9_H_11_)_2_]_2_, **4**, as cooperative photoredox catalysts in the oxidation of aromatic and aliphatic alkenes in water has been tested and the results have been studied for the purpose of comparison.

## 2. Results and Discussion

### 2.1. Synthesis, Spectroscopic, and Redox Characterization

The synthetic strategy followed for the preparation of the ruthenium-cobaltabis(dicarbollide) complex [Ru^II^(terpy)(bpy)(H_2_O)][(3,3′-Co(1,2-C_2_B_9_H_11_)_2_]_2_, **4** is described in [40,42] and the synthesis of [Ru^II^(terpy)(bpy)(H_2_O)][3,3′-Co(8,9,12-Cl_3_-1,2-C_2_B_9_H_8_)_2_]_2_ **5** involves the preparation of **Ag[Cl_6_-1]**, following the formation of **H[Cl_6_-1]** from the water insoluble **Cs [1]**. Then, cooperative system **5** is synthesized by dissolving [Ru^II^(terpy)(bpy)(Cl)]Cl, **2**, in a 1:1 mixture of water and acetone in the presence of **Ag[Cl_6_-1]** under reflux conditions, following the method described in [43]. After filtering out AgCl, a molecular aqua ruthenium (II) complex containing two hexachloro cobaltabis(dicarbollide) anions as counterions is isolated. The synthetic strategy is depicted in Figure 1.

Figure 1, Appendix A display the IR spectra of complexes **4**, **5**, and **Ag[Cl_6_-1]**. The IR of **4** and **5** show vibrations around 2900 and 3030 cm^−1^ that can be assigned to υ_C-H_ stretching modes for the aromatic rings of the cationic moieties and to the υ_C-H_ stretching of the C_c_-H bonds in the different rotamers of the dicarbollide anions, respectively. A band can be seen over 3500 cm^−1^ which corresponds to the υ_O-H_ stretching of the water ligands coordinated to ruthenium atoms in both compounds. We have also observed significant vibrations around 2530 cm^−1^ that correspond to υ_B-H_ stretching mode for the B-H bonds in the compounds. Unlike the spectrum of compound **4**, the spectra of compound **5** and **Ag[Cl_6_-1]** presents bands at 900 cm^−1^ corresponding to υ_B-Cl_ stretching of the B-Cl bonds, present in the hexachloro cobaltabis(dicarbollide) anions.

The 1D ^1^H NMR spectra of the complex **Ag[Cl_6_-1]** and compound **5**, along with the COSY spectrum of **5**, were recorded in acetone-*d*_6_. These are displayed in Figure 2, Appendix A, respectively. The ^1^H NMR spectrum of **5** shows two set of signals, similar to complex **4** [40], (i) one in the aliphatic region that can be assigned to the C_c_-H of the cobaltacarborane anions, which resonances appear around *δ* = 4.34 ppm, showing a shift of 0.36 ppm to low field with respect to complex **4**; and (ii) the second one, in the aromatic region corresponding to the protons of the bypyridine and terpyridine ligands of the ruthenium cation.

The ^1^H{^11^B} NMR spectrum exhibits the *H*-B resonances over a wide range of chemical-shift in the region from *δ* = 1.70 to 3.5 ppm (Appendix A). The ^11^B{^1^H} NMR spectra displays signals corresponding to the non-equivalent boron atoms avoiding the B-H signals coupling. On the other hand, the ^11^B NMR spectrum displays *B*-H signals as doublets and *B*-Cl signals as singlets. Both spectra show the typical pattern of the hexachloro cobaltabis(dicarbollide) cluster in the range from *δ* = 12.62 to −24.38 ppm, as shown in Appendix A [34].

The UV-vis spectrum of **Ag[Cl_6_-1]** in CH_2_Cl_2_ shows one strong absorption band at 327 nm and three others with less intensity at 280, 390, and 469 nm, (Appendix A) in accordance with [43,44,45]. Figure 3 shows the UV-visible spectra of **5** and **2**. The former exhibits ligand-based *π-π** bands of the cationic part below 350 nm that are partially eclipsed by strong absorptions corresponding to the hexachloro cobaltabis(dicarbollide) anionic moiety. Above 350 nm, the spectra show less-intense bands that correspond to d*π*(Ru)-*π**(L) MLCT transitions [46]. It is worth mentioning the shift to higher energy absorptions observed for the MLCT bands of **5** regarding those of complex **2**. This seems to indicate that the substitution of the chloride counterions by the metallacarboranes provokes the stabilization of the dπ (Ru) donor orbitals. 

The electrochemical behavior of complexes **Ag[Cl_6_-1]** and **5** have been studied by means of cyclic voltammetry (CV). The CV curve of **Ag[Cl_6_-1]** in CH_3_CN + 0.1 M [*n*-Bu_4_N][PF]_6_ (TBAH) shows one reversible one-electron-redox process at *E*_1/2_ = −0.89 V versus Ag/AgCl as reference electrode, which can be assigned to Co^III^/Co^II^ (see Appendix A). Unlike the derivative **Ag [1]**, the wave corresponding to Co^IV^/Co^III^ is not observed, probably because it is outside the working range of the solvent. 

The CV of complex **5** in CH_3_CN + 0.1 M TBAH exhibits different redox processes due to the ruthenium and cobalt ions. Two quasi-reversible monoelectronic Co^III^ Ru^III^/Co^III^ Ru^II^ and Co^III^ Ru^IV^/Co^III^ Ru^III^ redox waves at *E*_pa1_ = 0.82 V; *E*_pc1_ = 0.74 V and *E*_pa2_ = 1.17 V; *E*_pc2_ = 0.93 V vs. Ag/AgCl (*E*_pa1_ = 0.38 V; *E*_pc1_ = 0.31 V and *E*_pa2_ = 0.73 V; *E*_pc2_ = 0.49 vs. Fc^+^/Fc) respectively and one quasireversible monoelectronic Co^III^ Ru^II^/Co^II^ Ru^II^ redox wave at *E*_pa3_ = −0.76 V; *E*_pc3_ = −0.97 V vs. Ag/AgCl, (*E*_pa3_ = −1.19 V; *E*_pc3_ = −1.40 V vs. Fc^+^/Fc) (Appendix A). The redox wave corresponding to the Co^IV^ Ru^IV^/Co^III^ Ru^IV^ is not observed. The two successive one-electron oxidation waves that correspond to Co^III^ Ru^III^/Co^III^ Ru^II^ and Co^III^ Ru^IV^/Co^III^ Ru^III^ redox couples can be assigned to two PCETs as it was observed in complex **4 [40]**. These values have been assigned to the Ru^III^/Ru^II^ and Ru^IV^/Ru^III^ couples of the catalytic unit based in the values presented by the mononuclear [Ru^II^(terpy)(bpy)(H_2_O)]^2+^ compound, [42,47] and in accordance with the values observed in complex **4**. This last compound, unlike **5**, shows one more oxidation wave attributed to Co^IV^Ru^IV^/Co^III^Ru^IV^ redox couple a *E*_1/2_ = 1.38 vs. Ag/AgCl (0.95 V vs. Fc^+^/Fc). In **5**, this potential would likely be outside the working range due to the greater oxidizing character of the anion **[Cl_6_-1]^−^** compared to **[1]^−^**.

As has been observed with **4** in water [42], it is expected that in both compounds **4** and **5** the photogenerated Co^IV^ could oxidize [Ru^III^-OH]^2+^ to [Ru^IV^=O]^2+^, since the potential for Co^IV^/Co^III^ is high enough compared to the Ru^IV^/Ru^III^ couple. Then, it is expected that for photocatalytic oxidation of alkenes, both **[1]^−^** and **[Cl_6_-1]^−^** could act as good photosensitizers.

### 2.2. Photocatalytic Alkene Oxidations

The photocatalysts **4** and **5** are [Ru][Co]_2_ ion pairs based on Co, an abundant transition metal, whereas the redox parts are Ru-OH_2_ complexes. The first photocatalytic alkene oxidation experiments were all performed using **4** as the photocatalyst by exposing the reaction quartz vials to UV irradiation (2.2 W, λ ~ 300 nm) at room temperature and atmospheric pressure using styrene as substrate and very relevantly using water as a solvent. The samples were made of 5 mL of water pH = 7 (K_2_CO_3_ solution is used to adjust the pH) with a mixture of **4** (0.01 mM), styrene (20 mM), and Na_2_S_2_O_8_ (26 mM) as oxidizing agent, (catalyst load of 0.05 mol%). Then, different reaction times were tested to assess the conversion as a function of time. The reaction products were extracted with dichloromethane six times, dried with Na_2_SO_4_ and quantified by means of GC-MS analysis. As can be observed in Appendix A, after 30 min of reaction, even after 15 min, all the styrene was converted. Blank experiments after 30 min of reaction, in the absence of catalyst, light, or oxidizing agent, showed that no significant conversion of styrene occurred. Figure 4 displays (a) the yields of different oxidation products obtained at different times and (b) the molar fractions (X_M_) obtained in function of the reaction time. As we can see, after 5 min the amount of epoxide obtained is high (67%) and after 15 min the amount of diol increases, although the amount of epoxide is still greater (35% diol vs. 57% epoxide). A slight decrease in epoxide is observed after 30 min, (34% diol vs. 51% epoxide), but in any case, it continues to be the majority product above diol and other byproducts such as benzaldehyde and benzoic acid. We observed a slight decrease in yield and selectivity for epoxide formation with longer reaction times. Therefore, we selected a 15-min reaction time for the photoredox oxidation of various alkenes. However, we also examined the catalyst’s performance after a 30-min reaction in some instances.

Table 1 displays the performance of the photocatalyst **4** after 5, 15, and 30 min of reaction using 0.05 mol% and 0.005 mol% of catalyst. In general, when 0.05 mol% of catalyst is used, it can be observed that total conversion values were achieved after 15 min of reaction with, in many cases, total conversion after 30 min. However, moderate selectivity in the corresponding epoxide is obtained, which decreases when the reaction time increase, along with the TON values. However, with 0.005 mol% of catalyst the conversion values after 15 and 30 min are lower but the selectivity and TON values are higher. This happens simultaneously with an increase in the amount of the corresponding diol for most of the substrates studied (see Appendix A). At this point we can assert that the selectivity towards the epoxide increases with lower loads of catalyst **4**.

The photoepoxidation of *trans*-β-methylstyrene (entry 2), *trans*-stilbene (entry 3) led to the formation of the corresponding *trans*-isomer without isomerization after 15 and 30 min; however, in the case of *cis*-β-methylstyrene, the stereoselectivity towards the formation of the *cis*-epoxide isomer decreases after 30 min of reaction. The oxidation of aliphatic alkenes as the cyclic cyclooctene (entry 5) together with linear 1-octene (entry 6) has also been tested. In both cases, a good performance is shown. 

As we have commented previously, the selectivity towards the epoxide in water decreased with the reaction time whereas the amount of diol increases as well as, in some cases, other overoxidation products. In an aqueous medium, the epoxide ring undergoes ring opening. However, we aimed to determine whether this process is driven by water or by our catalyst. Different experiments were then performed by exposing the sealed reaction to UV irradiation for 15 and 30 min, operating with 0.05 mol% of catalyst, under the same conditions used before and using three different epoxides: 1,2-epoxyoctane, styrene oxide, and trans-stilbene oxide. The results shown in Appendix A indicate the formation of the corresponding diols in water after 15 and 30 min if the photocatalyst is present. When the catalyst is absent, lower or nonexistent yields of diols have been observed. In the case of styrene, only benzaldehyde of benzoic acid has been detected after 30 min. Thus, we can conclude that **4** acts as a photocatalyst in both processes, epoxidation, and hydroxylation in aqueous media. It is worth mentioning that ring opening of epoxides is a promising process to produce 1,2-diols, an important functional group to produce pharmaceuticals, surfactants, or their intermediates [48].

We have also studied the behavior of **5** as photocatalyst in the alkene oxidation in water using low catalyst load of 0.005 mol%. The results are presented in Table 2 and Appendix A, including those previously obtained for compound **4**, to facilitate a comparative analysis. In general, we can observe moderate-to-high conversion values after 30 min of reaction in most cases. It is worth noting that while the conversion values of catalyst **5** are generally higher than those obtained with **4**, in most of the substrates studied, the selectivity values in the corresponding epoxides are lower even after 15 min of reaction. This is probably due to the enhanced oxidizing capacity of Co^IV^ in catalyst **5**, as a consequence of the presence of more electron-withdrawing substituents on the cosane platform. At this point, we can assert that in most cases and under the studied conditions, photocatalyst **4** shows higher selectivity values towards the epoxidation of alkenes that photocatalyst **5**. 

Based on the photocatalytic results exposed above, we have postulated a mechanism, displayed in Figure 5, for the photoepoxidation of alkenes carried out by complexes **4** and **5**.

In the mechanism that is consistent with the products generated, the absorption of light by the photosensitizer Co_p_^III^ produces the excitation to form Co_p_^III^*, which undergoes oxidative quenching by S_2_O_8_^2−^ (oxidizing agent) generating Co_p_^IV^. This photogenerated strong oxidizing Co^IV^ can oxidize Ru_c_^II^-OH_2_ to Ru_c_^III^-OH. Then, the SO_4_^−^ radical oxidizes a new Co_p_^III^ to Co_p_^IV^ which oxidizes Ru_c_^III^-OH to Ru_c_^IV^=O species. The Ru_c_^IV^=O species reacts with the corresponding alkene to afford the oxidized products, with the regeneration of the corresponding catalyst Ru_c_^II^-OH_2_. With the pathway proposed in Figure 5, the exchange of two electrons and two protons takes place in the oxidation of alkenes.

## 3. Experimental

### 3.1. Materials, Instrumentation, and Measurements

The commercial Cs[Co(1,2-C_2_B_9_H_11_)_2_], **Cs [1]**, was obtained of Katchem Spol.sr.o. Compounds [Ru^III^Cl_3_(terpy)], [49], [Ru^II^Cl(terpy)(bpy)]Cl, **2**, [Ru^II^(terpy)(bpy)(OH)_2_](ClO_4_)_2_, **3** [42,47] and [Ru^II^(terpy)(bpy)(H_2_O)]**[(3,3′-Co(1,2-C_2_B_9_H_11_)_2_]_2_**, **4** [40] were also prepared according to literature procedures. All synthetic manipulations were performed under nitrogen atmosphere using vacuum line techniques.

All reagents used in the present work were obtained from Aldrich Chemical Co. and were used without further purification. Reagent grade organic solvents were obtained from SDS, and high purity deionized water was obtained by passing distilled water through a nano-pure Mili-Q water purification system. 

UV-*vis* spectroscopy was performed on a Cary 50 Scan Varian (Santa Clara, CA, USA) UV-*vis* spectrophotometer with 1 cm quartz cells or with an immersion probe of 5 mm path length. NMR spectra have been recorded with a Bruker ARX 300 instrument (Bruker Biospin, Rheinstetten, Germany) and ^1^H NMR spectra were recorded in acetone-d_6_. Chemical shift values were referenced to SiMe_4_. Elemental analyses were performed using a CHNS-O Elemental Analyser EA-1108 from Fisons (Waltham, USA). ESI-MS experiments were performed on a Navigator LC/MS chromatograph from Thermo Quest Finnigan (Toronto, Canada) using acetonitrile as mobile phase. Cyclic voltammetric (CV) or differential pulse voltammetry (DPV) was performed in an IJ-Cambria 660C potentiostat using a three-electrode cell. Glassy carbon electrode (3 mm diameter) from BAS was used as working electrode, platinum wire as auxiliary, and Ag as pseudo-reference electrode or SCE as the reference electrode. All cyclic voltammograms presented in this work were recorded under nitrogen atmosphere. The complexes were dissolved in deoxygenated solvents containing the necessary amount of TBAH as supporting electrolyte to yield a 0.1 M ionic strength solution. All *E*_1/2_ values reported in this work were estimated from cyclic voltammetry experiments as the average of the oxidative and reductive peak potentials (*E*_pa_ + *E*_pc_)/2. Unless explicitly mentioned, the concentration of the complexes was approximately 1mM.

Gas chromatography was performed with a GC-2010 Gas Chromatograph from Shimadzu (Kyoto, Japan), equipped with an Astec CHIRALDEX G-TA column, 30 m × 0.25 mm (i.d); FID detector, 250 °C; injection: 250 °C; carrier gas: helium; rate: 1.57 mL min^−1^; area normalization. The product analyses in the catalytic experiments were performed by GC with biphenyl as internal standard.

### 3.2. Synthesis of Compounds

Synthesis of Ag [3,3′-Co(8,9,12-Cl_3_-1,2-C_2_B_9_H_8_)_2_], **Ag[Cl_6_-1]**. A sample of Cs [3,3′-Co(1,2-C_2_B_9_H_11_)_2_], **Cs [1]** (0.45 g, 0.985 mmol) was dissolved in 14 mL of acetonitrile under magnetic stirring. It was then slowly added dropwise 14 mL of SO_2_Cl_2_ and the reaction mixture was refluxed 2 h at 70 °C. Afterwards, volatiles were removed in the rotavapor and the resulting residue was extracted three times with diethyl ether (15 mL) and aqueous HCl (3M, 3 × 15 mL) to remove all impurities. Then, the organic layer was dried over MgSO_4_. After filtration, the liquid was evaporated and dissolved in water and precipitated using a saturated aqueous solution of CsCl. An orange solid was isolated. Then, the solvent was evaporated and the remaining solid was dissolved in 20 mL of diethylether. Further purification was performed by extraction procedure by hydrochloric acid 1M (3 × 15 mL). The organic phase containing H [3,3′-Co(8,9,12-Cl_3_-1,2-C_2_B_9_H_8_)_2_] (**H[Cl_6_-1]**), was carried out to dryness under reduced pressure and dissolved with water. It was then added to an excess of silver nitrate (0.251 g, 1.477 mmol) to form a precipitate that was collected on a frit and washed with water three times and dried under vacuum. Yield 371.0 mg (60%). MALDI-TOF-MS: *m*/*z* calc. for [Cl_6_-1]^−^: 530.40; *m*/*z* found (%): 495.95 [Cl_5_-1] (8%), 529.91 [Cl_6_-1] (85.02%), 563.86 [Cl_7_-1] (7%). IR (*ν*, cm^−1^): 3049 (wk, C_c_-H), 2600 (shp, B-H), 1605 (O-H), 993 (B-Cl). ^1^H{^11^B} NMR (400 MHz, CD_3_OCD_3_): δ 4.35 (br s, 4H, C_c_-H), 3.51, 3.34, 3.08, 2.91, 2.32, 1.69 (br s, 12H, B-H). ^11^B NMR (96.29 MHz, CD_3_OCD_3_): δ = 12.14 (m, 2B, B-H, B8, B8′), 4.60 (m, 6B, B-H, B9, B9′, B12, B12′, B7, B7′), −5.24 (m, 4B, B-H, B10, B10′, B4, B4′), −17.90 (d, 1J(B,H) = 156 Hz, 4B, B-H, B5, B5′, B11, B11′), −24.38 (d, 1J(B,H) = 162 Hz, 2B, B-H, B6, B6′). E_1/2_ Co^III/II^(CH_3_CN + 0.1M TBAH): −0.89 V vs. Ag. UV-*vis* (CH_3_CN, 1 × 10^−4^ M): λ_max_ nm (ε, M^−1^ cm^−1^) 240 (2978), 281 (7264), 328 (29,289), 391 (4014), 472 (484).

Synthesis of [Ru^II^(terpy)(bpy)(H_2_O)][3,3′-Co(8,9,12-Cl_3_-1,2--C_2_B_9_H_8_)_2_]_2_, **5**. A sample of 0.09 g (0.142 mmol) of **2** and (0.26 g, 0.401 mmol) sample of **Ag[Cl_6_-1]** were dissolved in 60 mL of acetone:water (1:1) and the resulting solution was refluxed for 3 h. Then, AgCl was filtered off through a frit containing celite. The volume of the solution was reduced and the mixture chilled in a refrigerator for 48 h. The orange precipitate was collected on a frit, washed with cold water and anhydrous ethylether, and then vacuum-dried. Yield: 0.261 g (42%). Anal. Found (Calc.) for C_33_H_53_B_36_Cl_12_Co_2_N_5_ORu·0.77H_2_O·2Et_2_O: C, 28.52 (28.52); H, 3.71 (4.35); N, 3.90 (4.03) %. ^1^H NMR (acetone-*d*_6_, 400 MHz): δ 9.87 (d, 1H, ^3^*J_H-H_* = 5.6 Hz, 1H, H1), 8.97 (d, 1H, ^3^*J_H-H_* = 8.0 Hz, 1H, H7), 8.91 (d, ^3^*J_H-H_* = 7.9 Hz, 2H, H17, H19), 8.77 (d, ^3^*J_H-H_* = 8.2 Hz, 2H, H14, H22), 8.65 (d, ^3^*J_H-H_* = 8.1 Hz, 1H, H4), 8.57 (td, ^3^*J_H-H_* = 8.1 Hz, ^4^J*_H-H_* = 1.2 Hz, 1H, H8), 8.48 (t, ^3^*J_H-H_* = 8.3 Hz, 1H, H18), 8.19 (ddd, ^3^*J_H-H_* = 7.1 Hz, ^4^*J_H-H_* = 1.0 Hz, 1H, H9), 8.13 (td, ^3^*J_H-H_* = 7.9 Hz, ^4^*J_H-H_* = 1.2 Hz, 2H, H13, H23), 8.05 (d, ^3^*J_H-H_* = 5.4 Hz, 2H, H11, H25), 7.89 (td, ^3^*J_H-H_* = 8.4 Hz, ^4^*J_H-H_* = 1.3 Hz 1H, H3), 7.65 (m, 3H, H10, H12, H24), 7.57 (td, ^3^*J_H-H_* = 8.0Hz, ^4^*J_H-H_* = 1.3 Hz, 1H, H2), 5.83 (s, 2H, Ru-OH_2_), 4,34 (s, 8H, C_c_-H). ^1^H{^11^B} NMR (acetone-*d*_6_, 400 MHz): δ 4.34 (s, 8H, C_c_-H), 3.07 (s, 8B-H, B4, B4′, B7, B7′), 2.89 (s, 4B-H, B10, B10′), 1,97 (s, 4B-H, B6, B6′), 1.91 (s, 4B-H, B5, B5′), 1.67 (s, 4B-H, B11, B11′). ^11^B NMR (acetone-*d*_6_, 128 MHz): δ 12.59 (d, 4B, *J_B-H_* = 116.6 Hz, B-Cl), 4.82 (m, 8B, B-Cl), 1.99 (m, 4B, B-H), −5.17 (d, 4B, *J_B-H_* = 164.9 Hz, B-H), −17.94 (d, 8B, *J_B-H_* = 162.5 Hz, B-H), −24.41 (d, 4B, *J_B-H_* = 177.9 Hz, B-H). ^11^B{^1^H} NMR (acetone-*d*_6_, 128 MHz): δ 12.62 (brs, 4B, B8, B8′), 4.82 (s, 8B, B9, B9′, B12, B12′), 2.82 (s, 4B, B7, B7′), −5.05 (s, 4B, B10, B10′), −17.88 (s, 8B, B5, B5′, B11, B11′). **^1^**^3^C{^1^H}-NMR (acetone-*d*_6_, 100 MHz): δ 159.30, 159.14, 158.65, 156.02, 153.12, 152.74, 150.53, 138.45, 137.53, 136.19, 136.05, 127.96, 127.59, 126.46, 124.18, 124.12, 123.59 and 123.28, (C **ter**py-**Ru**-**b**py) and 47.42 (**C**_c_). E_1/2_ (CH_2_Cl_2_+ 0.1 M TBAH) Co^III/II^, −0.87 V; Co^IV/III^, 1.39; Ru^III/II^, 0.67 V; Ru^IV/III^, 0.97 V vs. Ag. IR (*ν_max_*, cm^−1^): 3052, 2924, 2853, 2583, 1694, 1602, 1446, 1465, 1447, 1386, 1100, 1025, 938, 858, 875, 760. UV-*vis* (CH_3_CN, 1 × 10^−5^ M) [λ_max_ nm (ε, cm^−1^ M^−1^)]:232 (24,156), 244 (21,064), 291 (40,756), 314 (57,401), 326 (60,042), 395 (8855), 464 (7124). MS (ESI^+^) in acetonitrile (*m*/*z*): 1591.3 [M + K]^+^, 1021.20 [M-**[Cl_6_-1]**^−^]^+^, 530.1 **[Cl_6_-1]**^−^.

### 3.3. Photocatalytic Studies

A quartz tube containing an aqueous solution (5 mL) at pH 7 (K_2_CO_3_ solution is used to adjust the pH) with **4** or **5**, as catalysts, alkene as substrate, and Na_2_S_2_O_8_ as sacrificial acceptor was exposed to UV light (2.2 W, λ = 300 nm or 352 nm) for different times. The complex: substrate: sacrificial oxidant ratios used (1:2000:2600 and 1:20,000:26,000 corresponding to concentrations of 0.01:20:26 mM and 0.001:20:26 mM, respectively). The concentrations were varied according to the study. For each experiment, a light reactor supplied light illumination with 12 lamps that produce UVA light at room temperature. The resulting solutions were extracted with CH_2_Cl_2_ six times. The solution was dried with anhydrous sodium sulfate and the solvent was reduced to a minimum volume under reduced pressure, then 100 µL of biphenyl 100 mM as internal standard was added to the resulting solution, 2 mM in the resulting 5 mL solution. To check the reproducibility of the reactions, all the experiments were performed in triplicate and analyzed by Gas Chromatography.

### 3.4. Gas Chromatography Studies

Studies were performed with a GC-2010 Gas Chromatograph from Shimadzu, equipped with an Astec CHIRALDEX G-TA column, 30 m × 0.25 mm (i.d) (Fukuoka, Japan); FID detector, 250 °C; injection: 250 °C; carrier gas: helium; rate: 1.57 mL/min; area normalization. For alkenes, substrates and products of catalysis were detected under the following conditions: styrene and derivatives: column temperature, 80 °C for 5 min, raising to 170 °C in a rate of 10 °C/min, holding 170 °C for 6 min. Trans-β-methyl-styrene, *cis*-β-methyl-styrene, *cis*-cyclooctene and derivatives: column temperature, 40 °C for 5 min, raising to 170 °C in a rate of 5 °C/min, holding 170 °C for 2 min. 1-octene and derivatives: column temperature, 30 °C for 5 min, raising to 170 °C in a rate of 10 °C/min, holding 170 °C for 3 min. Trans-stilbene and derivatives: column temperature, 50 °C for 1 min, raising to 150 °C in a rate of 15 °C/min, holding 150 °C for 2 min, then raising to 170 °C in a rate of 4 °C/min, holding 170 °C for 12 min. The product analyses in the catalytic experiments were performed by GC with biphenyl as internal standard.

## 4. Conclusions

Two cooperative ion pair photoredox systems [Ru^II^(terpy)(bpy)(H_2_O)]**[(3,3′-Co(1,2-C_2_B_9_H_11_)_2_]_2_**, **4** and [Ru^II^(terpy)(bpy)(H_2_O)][3,3′-Co(8,9,12-Cl_3_-1,2--C_2_B_9_H_8_)_2_]_2_ **5** have been studied as photoredox catalysts in the oxidation of aromatic and aliphatic alkenes in water. Both systems are formed by the same aqua ruthenium complex, [Ru^II^(terpy)(bpy)(OH_2_)]^2+^, as electron transfer agent and by cobaltabis(dicarbollides), as light collectors; in **4** by [3,3′-Co-(1,2-C_2_B_9_H_11_)_2_]^−^
**[1]^−^** and in **5** by [3,3′-Co(8,9,12-Cl_3_-1,2--C_2_B_9_H_8_)_2_]^−^ **[Cl_6_-1]**^−^. Both the cobaltabis(dicarbollide) and the ruthenium aqua complex are linked by non-covalent interactions, avoiding costly covalent bonding. Complex **4** was previously synthesized by us and **5** has been easily made by the reaction of the chlorido Ru(II) complex, with **Ag[Cl_6_-1]** in water/acetone (1:1) under reflux. The electrochemical studies evidence that the photogenerated Co^IV^ can easily oxidize [Ru^III^-OH]^2+^ to [Ru^IV^=O]^2+^ in water via PCETs as we have evidenced in the photocatalytic oxidation of alkenes in water.

We have highlighted the capacity of **4** to perform as excellent cooperative photoredox catalyst in the oxidation of alkenes in water using catalyst loads of 0.05 and 0.005 mol%, achieving high yields even when short reaction times of irradiation have been used. Using 0.05 mol% of catalyst, the epoxidation of styrene led to a slight decrease in epoxide after 30 min, but in any case, it continues to be the majority product above diol and other byproducts such as benzaldehyde and benzoic acid. For the rest of the substrates studied, the selectivity of epoxide decreases after 30 min. With 0.005 mol% of photocatalyst, the conversion values after 15 and 30 min are lower but the selectivity for the corresponding epoxide and TON values are higher. Compound **4** serves as a photocatalyst for both epoxidation and hydroxylation processes in aqueous media. When using catalyst **5** with a 0.005 mol% loading, the conversion values are typically higher than those achieved with compound **4**. However, in most cases, the selectivity towards the corresponding epoxides is lower, even after a 15-min reaction period. This difference is likely attributed to the enhanced oxidizing capacity of Co^IV^ in catalyst **5**.

To our knowledge, these are the first examples of cooperative systems featuring robust non-bonding interactions, which omit covalent bonds and have not previously been explored as photoredox catalysts for the epoxidation of alkenes in water. We have proposed a potential mechanism. 

## Data Availability

Data are contained within the article and Appendix A.

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
