# Peer review of "Unveiling Non-Covalent Interactions in Novel Cooperative Photoredox Systems for Efficient Alkene Oxidation in Water"

_molecules, 2024, doi:10.3390/molecules29102378_

Round 1

Reviewer 1 Report

Comments and Suggestions for Authors

This manuscript by I. Romero, F. Teixidor and coworkers reports the synthesis of a new ion pair [Ru][Co]2 complex (5) and study this complex together with a related previously reported complex (4; Inorg. Chem. 2023) in photocatalytic oxidation of olefins to epoxides. The photocatalysis is performed in an aqueous medium and the authors use relatively low catalyst loadings, 0.05 and 0.005 mol%. The authors found that 5 is more active than 4, and relate this with the higher oxidizing capacity of 5. The authors have previously reported related reactions of photooxidation reactions of alcohols and olefins using similar [Ru][Co]2 ion pair complexes, which impact on the novelty of the study. Overall the manuscript is well written and the subject is introduced appropriately. Therefore, I support the publication of the manuscript after the following concerns have been addressed:

- The cyclic voltammetry experiments in organic solvents are reported respect to Ag/AgCl. They should be referenced respect to ferrocene/ferrocenium following the IUPAC recommendation of using ferrocene as an internal reference redox system in non-aqueous media and that of experts in the matter (Neil G. Connelly and William E. Geiger, Chem. Rev. 1996, 96, 877-910 877).

- The CVs do not display starting point of the scan and the direction of the scan as is now established in most guidelines for reporting electrochemical data in journals. Equally they do not report the scan rate at which they were measured.

- Given that the photocatalysis is performed in an aqueous medium, the CV of 5 should be measured in aqueous solution is order to compare better to the reaction conditions.

- In pg.7, line 184; the redox waves are described as “almost reversible”. This is not a correct terminology for the description of a cyclic voltammetry wave which can be reversible, irreversible, or quasi-reversible. In order to establish this the redox process needs to be measured at different scan rates in order to observe the return current, something not reported by the authors. Looking at the CVs the waves looks more like irreversible. E1/2 can only be used for fully reversible systems. For quasi-reversible or irreversible systems, the anodic peak potential (Ep,a) and cathodic peak potential (Ep,c) should be used instead. The authors should address this.

A few minor corrections:

- In Table 1 and Table 2, substrate and product should be capitalized

- In Figure 2, The NMR spectra is cut off and should be corrected

Author Response

Reviewer 1:

1) The cyclic voltammetry experiments in organic solvents are reported respect to Ag/AgCl. They should be referenced respect to ferrocene/ferrocenium following the IUPAC recommendation of using ferrocene as an internal reference redox system in non-aqueous media and that of experts in the matter (Neil G. Connelly and William E. Geiger, Chem. Rev. 1996, 96, 877-910 877).

All potentials are now referenced to ferrocene/ferrocenium according with the reviewer’s suggestion

2) The CVs do not display starting point of the scan and the direction of the scan as is now established in most guidelines for reporting electrochemical data in journals. Equally they do not report the scan rate at which they were measured.

Now the CVs display starting point of the scan and the scan rate has been reported in the caption of Figure S6.

3) Given that the photocatalysis is performed in an aqueous medium, the CV of 5 should be measured in aqueous solution is order to compare better to the reaction conditions.

The electrochemical transition of [Cl6-1] from Co(III) to Co(IV) occurs beyond the electrochemical window of water in a buffer, rendering it unobservable. Conversely, a cyclic voltammetry (CV) analysis was conducted on complex 5 in buffered water, revealing all characteristic features of the complex. However, despite the fidelity of the observations it did not meet the standards for inclusion in the main manuscript due to its lack of aesthetic appeal. So we decided to relegate it to the Supplementary Information. Regrettably, further experimentation is unfeasible as the PhD student responsible for its synthesis has concluded their tenure, and we lack the resources to replicate the complex in a timely manner.

4) In pg.7, line 184; the redox waves are described as “almost reversible”. This is not a correct terminology for the description of a cyclic voltammetry wave which can be reversible, irreversible, or quasi-reversible. In order to establish this the redox process needs to be measured at different scan rates in order to observe the return current, something not reported by the authors. Looking at the CVs the waves looks more like irreversible. E1/2 can only be used for fully reversible systems. For quasi-reversible or irreversible systems, the anodic peak potential (Ep,a) and cathodic peak potential (Ep,c) should be used instead. The authors should address this.

We agree with the reviewer comment and we have corrected it in the text. We have written: “Two quasireversible monoelectronic CoIII RuIII/ CoIII RuII and CoIII RuIV/ CoIII RuIII redox waves at Epa1= 0.82 V; Epc1= 0.74V and Epa2= 1.17 V; Epc2= 0.93V vs Ag/AgCl (Epa1= 0.38 V; Epc1= 0.31V and Epa2= 0.73 V; Epc2= 0.49 vs Fc+/Fc) respectively and one quasireversible monoelectronic CoIII RuII /CoII RuII redox wave at Epa3= -0.76 V; Epc3=-0.97V vs Ag/AgCl, (Epa3= -1.19 V; Epc3=-1.40V vs Fc+/Fc).”

A few minor corrections:

4) In Table 1 and Table 2, substrate and product should be capitalized

We agree with the reviewer comment and we have corrected it in the text.

5 )In Figure 2, The NMR spectra is cut off and should be corrected

We agree with the reviewer comment and we have corrected it in the text.

We thank reviewer 1 for his/her comments that we hope we have addressed adequately.

Reviewer 2 Report

Comments and Suggestions for Authors

1. In Fig. 1, I suggest the authors have to label the main peaks, which will be easy for the reader.

2. I can not find the Fig. 2 well, pls list it well position.

3. Fig. 3 should be also marked.

4. The authors have given the proposed mechanism for alkene photooxidation, but I can not find any supporting information.

5. Pls illustrate the recycle number, it is practical application.

6. Work on such topic may be addressed, such as Molecules 2023, 28, 4507.

Author Response

Reviewer 2:

1) In Fig. 1, I suggest the authors have to label the main peaks, which will be easy for the reader.

We agree with the reviewer comment and we have added it in Fig.1.

2) I cannot find the Fig. 2 well, pls list it well position.

We agree with the reviewer and we have corrected it in the text.

3) Fig. 3 should be also marked.

We agree with the reviewer comment and we have added it in Fig.3.

4) The authors have given the proposed mechanism for alkene photooxidation, but I can not find any supporting information.

We consider that the proposed pathway is consistent with the experimental data generated and for the moment the one that explains best the results obtained. However, and considering the opinion of the reviewer we have specified that the pathway is postulated.

5) Pls illustrate the recycle number, it is practical application.

The studies that we show in our work are catalytic studies in homogeneous phase, so we have only made one cycle and have not carried out recycling, although in previous works we did recover the photocatalyst that had been previously heterogenized thanks to the strong interactions of the metallacarborane with amine residues (ACS Appl. Mater. Interfaces. 2020, 12, 56372-56384)

6) Work on such topic may be addressed, such as Molecules 2023, 28, 4507).

The reference “[[1]4]. Ye, D.; Liu, L ; Peng Q. ; Qiu, J. ; Gong H. ; Liu S. Effect of Controlling Thiophene Rings on D-A Polymer Photocatalysts Accessed via Direct Arylation for Hydrogen Production. Molecules 2023, 28, 4507”. has been added.

We thank reviewer 2 for his/her comments on the manuscript.

Reviewer 3 Report

Comments and Suggestions for Authors

Guerrero et al. continued the development of their compound 4 (ref. [43]) that they used for oxidation of alcohols, but this time for alkenes oxidation (epoxides formation). They also synthesized a new chloro derivative (5) and compared their catalytic properties. They also proved that the formation of diol byproduct is due to the catalyst, not just water. The article is quite well written (with strange variations of font/size everywhere).

My major concern is about the pH of the reaction mixture. For example, the authors stated that they used “5 ml aqueous K2CO3 solution at pH=7” (stated in several places in main text and SI). First, there is no information about the concentration in K2CO3 of this solution (saturated?). Moreover, for this reaction that involves two protons and is expected to be pH dependent, they do not use buffer but K2CO3 that is a base, not a buffer. Thus, it seems that it is impossible to have a neutral pH just using a base like K2CO3. Before acceptance, this must be clarified for the reaction conditions to be fully described and for reproducibility.

The authors fully described the state of art but with a quite excessive use of self-citation (20 on 55).

 Below are my raw comments:

Line 6 "are not linked by covalent bonds, avoiding covalent bonding”: It is a pleonasm, written like that.

There are variations in the font size in the document, for example in lines 92, 104, 109, 319, 353 ...

Line 10 “fully characterized by the first time.”: for the first time.

Self-citation is quite high (20/55: [11,12,13, 16, 17, 25, 26, 27,32, 33, 34, 35, 36, 38, 40, 41, 42, 43, 44, 47]).

Line 99 Chart 1: It is better to write the compound numbers (4 and 5) under the structure, even if stated in the caption. 5 has the oxidation degree of Ru, not 4 (the same Scheme 1). Since the Ru part is the same for both 4 and 5, why not doing a copy/past, to have identical patterns? The “2” is not in the same size/font.

Line 105 “…the synthesis of [RuII(terpy)(bpy)(H2O)][3,3’-Co(8,9,12-Cl3-1,2-C2B9H8)2]2 5 involves the preparation of Ag[Cl6-1], following the formation of H[Cl6-1] from the water insoluble Cs[1].”: In fact, Cs[Cl6-1] (in scheme 1), but that was done starting from Cs[1]. Since this step does not appear in scheme 1, this is quite confusing. We must wait for line 384 to understand this process.

Scheme 1:

- No need of bold for Et3N.

- For the preparation of 2, Ru is +II (oxidation degree, not formal charge) but +III for its precursor, without the use of redox reagents above the arrow (the same in ref. [43]). Please confirm that it is correct.

- According to line 352, “1” should be written under the structure of the precursor. But this number is already used for the anionic cobaltabis(dicarbollide) (in the form Cs[1], Ag[Cl6-1], H[Cl6-1]). This is confusing. Please check.

- The synthesis of Ag[Cl6-1], done in this work starting from Cs[1] and described line 384 should appear in scheme 1.

Figure 2: Remark: The spectrum is cut in my PDF (end of page problem?). But is OK in Figure S4.

178 “+ 0.1 M TBAH”: Describe acronyms the first time they appear. This description only appear line 373 at the third occurrence.

209, 252 and 436 “(K2CO3 pH=7)”: I do not understand how it is possible to have neutral pH with a base like K2CO3. Did you mean KHCO3? Please, control pH. For line 436, the 3 is not in subscript.

Figure 4: a) "t (min)" but for b) "t(min)".

Table 1 “tcis” for entry 4.

Figure 5 is very similar to figure 5 of ref. [43] but without the second irradiation (at 7 o’clock if comparing the cycle to a clock). Is there a reason for this difference?

351 “The commercial Cs[Co(1,2-C2B9H11)2], Cs[1], was obtained of Katchem Spol.sr.o. Compounds [RuIIICl3(terpy)], 1 [53],”: See above for the double use of number 1.

391 “An orange solid was isolated.”: Is it Cs[Cl6-1], appearing in scheme 1?

References

[17] volume not in italic.

SI:

Caption for Figure S7 (both in table of contents and above Figure S7) “…4(0.01 mM), styrene (20 mM),…”: Missing space after 4, that should be in bold.

TableS2. Photooxidation of epoxides” (in table of contents): Missing space.

In table of contents:

Figure S6. CV of a) Ag[Cl6-1] compound in CH3CN + 0.1 M TBAH vs Ag/AgCl; and b) 5 in CH3CN + 0.1 M TBAH vs Ag/AgCl.

But above Figure S6 (“5-Cl6” to be corrected):

Figure S6. CV of a) Ag[Cl6-1] compound in CH3CN + 0.1 M TBAH vs Ag/AgCl; and b) 5-Cl6. in CH3CN + 0.1 M TBAH vs Ag/AgCl.

Author Response

Reviewer 3:

1) My major concern is about the pH of the reaction mixture. For example, the authors stated that they used “5 ml aqueous K2CO3 solution at pH=7” (stated in several places in main text and SI). First, there is no information about the concentration in K2CO3 of this solution (saturated?). Moreover, for this reaction that involves two protons and is expected to be pH dependent, they do not use buffer but K2CO3 that is a base, not a buffer. Thus, it seems that it is impossible to have a neutral pH just using a base like K2CO3. Before acceptance, this must be clarified for the reaction conditions to be fully described and for reproducibility.

We agree with the referee's comment, and therefore this has been corrected in the text. The reaction medium is water, and to bring this solution to neutral pH, K2CO3 solution is added. In the catalytic experiments, the pH is controlled and adjusted to 7 when it is observed that it has decreased.

 Below are my raw comments:

2) Line 6 "are not linked by covalent bonds, avoiding covalent bonding”: It is a pleonasm, written like that.

We agree with the reviewer’s comment and we have written: “photocatalysts in which the catalyst and photosensitizer are not linked by covalent bonds.”

3) There are variations in the font size in the document, for example in lines 92, 104, 109, 319, 353 ...

We have corrected it in the text.

4) Line 10 “fully characterized by the first time.”: for the first time.

We have corrected it in the text.

5) Self-citation is quite high (20/55: [11,12,13, 16, 17, 25, 26, 27,32, 33, 34, 35, 36, 38, 40, 41, 42, 43, 44, 47]).

The referee correctly pointed out that the references cited in the article are relevant due to the group's extensive research on cobaltabis(dicarbollide) and its derivatives over many years. These references encompass previous catalysis results and cobaltabis properties, crucial for understanding the article's content. However, upon further review, references 12, 27 and 34 have been removed.

6) Line 99 Chart 1: It is better to write the compound numbers (4 and 5) under the structure, even if stated in the caption. 5 has the oxidation degree of Ru, not 4 (the same Scheme 1). Since the Ru part is the same for both 4 and 5, why not doing a copy/past, to have identical patterns? The “2” is not in the same size/font.

We have corrected the Chart 1 according to the reviewer’s suggestion.

7) Line 105 “…the synthesis of [RuII(terpy)(bpy)(H2O)][3,3’-Co(8,9,12-Cl3-1,2-C2B9H8)2]2 5 involves the preparation of Ag[Cl6-1], following the formation of H[Cl6-1] from the water insoluble Cs[1].”: In fact, Cs[Cl6-1] (in scheme 1), but that was done starting from Cs[1]. Since this step does not appear in scheme 1, this is quite confusing. We must wait for line 384 to understand this process.

We have corrected the Scheme 1 for better understanding.

Scheme 1:

8) No need of bold for Et3N.

We have corrected it in Scheme 1.

9) For the preparation of 2, Ru is +II (oxidation degree, not formal charge) but +III for its precursor, without the use of redox reagents above the arrow (the same in ref. [43]). Please confirm that it is correct.

Reaction of equimolar amount of [RuIIICl3(terpy)] (starting material) and the ligand 2,2′-bipyridine (bpy) in EtOH:H2O at reflux in the presence of Et3N resulted in the formation of the chlorido RuII complex 2. Triethylamine acts as a reducing agent in this medium, leading to the reduction of RuIII to RuII.

10)According to line 352, “1” should be written under the structure of the precursor. But this number is already used for the anionic cobaltabis(dicarbollide) (in the form Cs[1], Ag[Cl6-1], H[Cl6-1]). This is confusing. Please check.

We agree with the reviewer’s comment and we have removed 1 of our precursor.

-11)The synthesis of Ag[Cl6-1], done in this work starting from Cs[1] and described line 384 should appear in scheme 1.

We agree with the reviewer comment and we have added it to Scheme 1.

12) Figure 2: Remark: The spectrum is cut in my PDF (end of page problem?). But is OK in Figure S4.

We agree with the reviewer comment and we have corrected it in the text.

13)178 “+ 0.1 M TBAH”: Describe acronyms the first time they appear. This description only appear line 373 at the third occurrence.

We agree with the reviewer's comment and we have added the acronym description the first time it appeared, that is, on page 178.

14) 209, 252 and 436 “(K2CO3 pH=7)”: I do not understand how it is possible to have neutral pH with a base like K2CO3. Did you mean KHCO3? Please, control pH. For line 436, the 3 is not in subscript.

As we have previously mentioned, we agree with the referee's comment, and therefore this has been corrected in the text. The reaction medium is water, and to bring this solution to neutral pH, K2CO3 solution is added. In the catalytic experiments, the pH is controlled and adjusted to 7 when it is observed that it has decreased.

15) Figure 4: a) "t (min)" but for b) "t(min)".

We have corrected it in Figure 4.

16)Table 1 “tcis” for entry 4.

we have corrected this mistake

17) Figure 5 is very similar to figure 5 of ref. [43] but without the second irradiation (at 7 o’clock if comparing the cycle to a clock). Is there a reason for this difference?

There is no reason for this difference, both proposals are correct, in any case the two proposed oxidation processes take place in the presence of light.

18) 351 “The commercial Cs[Co(1,2-C2B9H11)2], Cs[1], was obtained of Katchem Spol.sr.o. Compounds [RuIIICl3(terpy)], 1 [53],”: See above for the double use of number 1.

We agree with the reviewer’s comment and we have removed 1 of the [RuIIICl3(terpy)] compound.

19) 391 “An orange solid was isolated.”: Is it Cs[Cl6-1], appearing in scheme 1?

yes that's how it is

20) References

[17] volume not in italic.

Ok , it has been changed.

21) SI:

Caption for Figure S7 (both in table of contents and above Figure S7) “…4(0.01 mM), styrene (20 mM),…”: Missing space after 4, that should be in bold.

TableS2. Photooxidation of epoxides” (in table of contents): Missing space.

In table of contents:

Figure S6. CV of a) Ag[Cl6-1] compound in CH3CN + 0.1 M TBAH vs Ag/AgCl; and b) in CH3CN + 0.1 M TBAH vs Ag/AgCl.

But above Figure S6 (“5-Cl6” to be corrected):

Figure S6. CV of a) Ag[Cl6-1] compound in CH3CN + 0.1 M TBAH vs Ag/AgCl; and b) 5-Cl6in CH3CN + 0.1 M TBAH vs Ag/AgCl.

The SI has been corrected according to the reviewer's suggestions.

We thank reviewer 3 for his/her comments on the manuscript.
